# Resting-State Isolated Effective Connectivity of the Cingulate Cortex as a Neurophysiological Biomarker in Patients with Severe Treatment-Resistant Schizophrenia

**DOI:** 10.3390/jpm10030089

**Published:** 2020-08-14

**Authors:** Masataka Wada, Shinichiro Nakajima, Ryosuke Tarumi, Fumi Masuda, Takahiro Miyazaki, Sakiko Tsugawa, Kamiyu Ogyu, Shiori Honda, Karin Matsushita, Yudai Kikuchi, Shinya Fujii, Daniel M. Blumberger, Zafiris J. Daskalakis, Masaru Mimura, Yoshihiro Noda

**Affiliations:** 1Department of Neuropsychiatry, Keio University School of Medicine, Tokyo 160-8582, Japan; m.wada@keio.jp (M.W.); ryousuke1114@gmail.com (R.T.); fumi_masuda@keio.jp (F.M.); takahime.miyazaki@nifty.com (T.M.); sakiko.tsugawa@gmail.com (S.T.); camille.1896@gmail.com (K.O.); mimura@a7.keio.jp (M.M.); 2Department of Psychiatry, Komagino Hospital, Tokyo 193-8505, Japan; 3Graduate School of Media and Governance, Keio University, Kanagawa, Tokyo 252-0882, Japan; shiori.0913.honda@keio.jp; 4Faculty of Environment and Information Studies, Keio University, Kanagawa, Tokyo 252-0882, Japan; t17752km@sfc.keio.ac.jp (K.M.); yudai-kikuchi@keio.jp (Y.K.); fujii.shinya@gmail.com (S.F.); 5Temerty Centre for Therapeutic Brain Intervention, Centre for Addiction and Mental Health, Department of Psychiatry, University of Toronto, Toronto, ON M6J 1H4, Canada; Daniel.Blumberger@camh.ca (D.M.B.); Jeff.Daskalakis@camh.ca (Z.J.D.)

**Keywords:** treatment-resistant schizophrenia, causal effective connectivity, isolated effective coherence, resting-state electroencephalography, anterior cingulate cortex, posterior cingulate cortex, default mode network

## Abstract

**Background**: The neural basis of treatment-resistant schizophrenia (TRS) remains unclear. Previous neuroimaging studies suggest that aberrant connectivity between the anterior cingulate cortex (ACC) and default mode network (DMN) may play a key role in the pathophysiology of TRS. Thus, we aimed to examine the connectivity between the ACC and posterior cingulate cortex (PCC), a hub of the DMN, computing isolated effective coherence (iCoh), which represents causal effective connectivity. **Methods**: Resting-state electroencephalogram with 19 channels was acquired from seventeen patients with TRS and thirty patients with non-TRS (nTRS). The iCoh values between the PCC and ACC were calculated using sLORETA software. We conducted four-way analyses of variance (ANOVAs) for iCoh values with group as a between-subject factor and frequency, directionality, and laterality as within-subject factors and post-hoc independent *t*-tests. **Results**: The ANOVA and post-hoc *t*-tests for the iCoh ratio of directionality from PCC to ACC showed significant findings in delta (*t*_45_ = 7.659, *p* = 0.008) and theta (*t*_45_ = 8.066, *p* = 0.007) bands in the left side (TRS
< nTRS). **Conclusion**: Left delta and theta PCC and ACC iCoh ratio may represent a neurophysiological basis of TRS. Given the preliminary nature of this study, these results warrant further study to confirm the importance of iCoh as a clinical indicator for treatment-resistance.

## 1. Introduction

Approximately one-third of patients with schizophrenia do not respond to antipsychotic treatment [1,2], which is considered as treatment-resistant schizophrenia (TRS). As the quality of life in patients with TRS is remarkably disturbed through their lifespan, understanding the pathophysiology of TRS is a priority for mental health research. However, the neural basis of TRS, especially the difference from non-treatment resistant schizophrenia (nTRS), remains unclear [3].

One brain region commonly reported to show abnormal structural and functional findings in patients with schizophrenia is the anterior cingulate cortex (ACC) [4,5,6]. The ACC is an area crucial for integrating emotional, cognitive/attentional, and nociceptive functioning, as well as motor processing [7]. Additionally, previous proton magnetic resonance spectroscopy studies demonstrated that patients with TRS showed increased levels of glutamatergic neurometabolites in the ACC compared with patients with nTRS [8] or healthy controls [9,10,11]. Thus, while dysfunction of the ACC is among pathological neural bases for schizophrenia, it may also be related to that for TRS.

Several resting-state functional magnetic resonance imaging (fMRI) studies have shown that connectivity within the default mode network (DMN) is increased in patients with schizophrenia compared with healthy controls [12,13,14]. The DMN correlates closely with the resting-state human brain activity and is thought to be involved in the monitoring of internal processes as well as internal and external cognition [15]. A number of studies suggested that impaired DMN may be related to various types of symptoms such as cognitive impairment and psychotic symptoms and be associated with long-term clinical outcomes in patients with schizophrenia [16,17,18]. In addition, previous fMRI studies have indicated that connectivity between the posterior cingulate cortex (PCC), one of the core nodes of the DMN, and ACC is associated with both positive and negative symptoms in patients with schizophrenia [19,20,21]. Notably, Alonso-Solís et al. reported that patients with TRS demonstrated decreased functional connectivity between the PCC and ACC compared with patients with nTRS [22]. Moreover, patients with schizophrenia who had higher severity of hallucination or delusions demonstrated reduced fractional anisotropy values of the cingulum bundle, as measured by diffusion tensor tractography [23], as well as a reduced magnetization transfer ratio, as measured by MRI [24]. These findings suggest that both functional and structural aberrant connectivity between the ACC and DMN may play a key role in the pathophysiology of TRS.

A recent development in computational techniques has enabled non-invasive measurements of scalp electroencephalography (EEG) to estimate not only local activities at arbitrary brain regions, but also functional connectivities between any two brain regions. Recently, in particular, a new method has been developed to calculate effective directional connectivities called “isolated effective coherence (iCoh)” [25]. The iCoh is considered to represent one of the causal effective connectivities that can specifically estimate the directionality of information flow along a specific path. Most of the brain nodes not only directly, but also indirectly affect one another. Distinguishing between them leads to more precise information. Although it is difficult to do so, the partial directed coherence (PDC) can be used to quantify direct connections that are not confounded by indirect paths, their directionality, and their spectral characteristics. However, this method is influenced by the sender nodes of interest and may decrease in the presence of many nodes, even if the relationship between a sender and receiver node of a particular interest remains unchanged [26]. Here, the iCoh is a novel method that overcomes the abovementioned limitations by estimating the partial coherence under a multivariate autoregressive model. Of note, the better accuracy of the iCoh method has been confirmed by several studies compared with the PDC [25,26,27].

For further investigation of the pathophysiology of TRS, it is indispensable to detect the direction of abnormality between the ACC and DMN. In this study, we hypothesized that the aberrant effective connectivity between the PCC and ACC may be associated with the pathophysiology of TRS. Therefore, we aimed to investigate the causal effective connectivities as indexed by iCoh of resting-state EEG, focusing on the path between the PCC and ACC between patients with TRS and nTRS.

## 2. Materials and Methods

### 2.1. Participants

This study was approved by the ethics committees at Komagino Hospital (IRB code: 20160504) on 22 October 2016 and Keio University School of Medicine (IRB code: 20160320) on 23 July 2018. All participants were included following the completion of an informed consent procedure. All patients were recruited from Komagino Hospital, Tokyo, Japan and had a diagnosis of schizophrenia or schizoaffective disorder based on the Diagnostic and Statistical Manual of Mental Disorders IV. Seventeen patients with TRS and thirty patients with nTRS were enrolled in this study. Treatment-resistance to antipsychotics was defined by the modified treatment response and resistance in psychosis (TRRIP) working group consensus criteria [28]. Specifically, TRS criteria included a history of treatment failure to optimal treatment with at least two previous non-clozapine antipsychotics, while nTRS criteria included the following: (i) current intake of a non-clozapine single antipsychotic and (ii) treatment response: every positive and negative syndrome scale (PANSS) [29] positive score less than 3 points, and clinical global impression score less than 3 points. We excluded patients who had (i) substance abuse/dependence within the past 6 months; (ii) history of head trauma resulting in loss of consciousness for more than 30 min; (iii) serious or unstable physical illness; or (iv) current administration of lamotrigine, topiramate, or memantine.

### 2.2. Clinical Assessments

The severity of clinical symptoms was assessed with the PANSS by experienced qualified psychiatrists (R.T. and S.N.).

### 2.3. Measurement and Preprocessing of Resting-State EEG

Resting-state EEG was acquired for approximately 5 min with a 19-channel EEG system (Neurofax EEG-1214, Nihon Kohden, Inc., Tokyo, Japan) according to the 10–20 international system using a linked earlobes reference. Subjects were instructed to keep their eyes closed while staying awake during the EEG recording. EEG data were recorded at the sampling rate of 500 Hz and electrode impedances were kept below 5 kΩ during the recording. EEG data were band-pass filtered off-line at 0.1–100 Hz. Blink and eye-movement related artifacts were removed using independent component analysis. After removing the periods contaminated with noise with a visual inspection, EEG data were concatenated and preprocessed with R software (2018). Subsequently, preprocessed EEG data was processed using standardized low-resolution brain electromagnetic tomography, which is implemented within sLORETA software [25,30].

### 2.4. iCoh Analysis

In the present study, we calculated the causal effective connectivity as indexed with the iCoh using sLORETA software [25] among the various functional connectivity indices. The iCoh is defined by the formula based on a multivariate autoregressive model, calculating the corresponding partial coherences after setting all irrelevant connections to zero other than the particular directional paths of interest. Here, a multivariate autoregressive model is a mathematical model of two-time series data that can be estimated using a linear sum of the history of the two-time series data. The partial coherence is a measure of connection between two complex-valued random variables after removing the effect of other measured variables. Again, technical details are described in a previously published article [25]. Information on effective connectivity provided by the iCoh method is supposed to represent “direct” paths of connections between the pairs of regions, excluding the influence of indirect connection paths [25]. Furthermore, iCoh provides two-directional estimators for the strength of oscillatory information flow between each pair of regions such as from region “A” to “B” and from region “B” to “A” [31].

The primary analysis of causal effective connectivity as indexed by iCoh was performed for region of interest (ROI) pairs between the PCC and ACC individually for each group. Subsequently, connectivity for each frequency band (i.e., delta: 1.5–3 Hz, theta: 4–7 Hz, alpha: 8–13 Hz, beta: 14–30 Hz, low-gamma: 30–45 Hz, and high-gamma: 55–70 Hz) was calculated. The ROI names, abbreviations, and the Montreal Neurological Institute (MNI)-coordinates are listed in Appendix A.

### 2.5. Statistical Analysis

Statistical analyses were performed using the SPSS software (version 25, SPSS Inc., Chicago, IL, USA). Clinico-demographic characteristics, including age, sex, years of education, age of onset, treatment duration, chlorpromazine (CPZ), and PANSS total scores were compared between the groups by χ^2^-tests or independent *t*-tests for categorical or continuous variables, respectively. In this study, normal distributions of the iCoh data were confirmed with Shapiro–Wilk tests before performing the parametric statistical testing. The iCoh values were statistically analyzed by four-way repeated-measures analysis of variance (rm-ANOVA) using “group” (i.e., two groups: TRS and nTRS) as a between-subject factor and “frequency” (i.e., six frequency bands: delta, theta, alpha, beta, low-gamma, and high-gamma), “directionality” (two directions: e.g., PCC to ACC and ACC to PCC), and “laterality” (two lateralities: right and left) as within-subject factors. When significant differences were found in any interactions, subsequent post-hoc rm-ANOVAs (i.e., three-way and two-way ANOVAs) were conducted. Finally, we performed post-hoc independent *t*-tests for the ratio of bidirectionality of iCoh values. The ratio was calculated as follows: [(PCC to ACC) − (ACC to PCC)]/[(PCC to ACC) + (ACC to PCC)]. Here, the significance level of alpha was set as 0.05, however, only for post hoc analyses of four-way ANOVA, the alpha level was set as 0.01 depending on the number of frequency bands (0.05/5 = 0.01). Pearson’s correlation coefficients between chlorpromazine (CPZ) equivalent daily doses and iCoh values were calculated in order to check the effect of antipsychotics on the iCoh as a confounding factor.

In addition, Pearson’s correlation coefficients were calculated for the results showing significant findings in the above ANOVA model to examine the correlations among iCoh values within the ROIs and clinical symptoms as assessed with the PANSS total scores.

Moreover, we conducted a receiver operating characteristic (ROC) analysis to investigate the sensitivity and specificity of the iCoh index in discriminating between TRS and nTRS.

## 3. Results

### 3.1. Clinico-Demographics Data

Clinico-demographic data are summarized in Table 1. There were no significant group differences in age, sex, years of education, age of onset, and treatment duration other than CPZ equivalent daily doses and PANSS total scores, suggesting the nature of differences between TRS and nTRS. 

### 3.2. Four-Way ANOVA for iCoh Values

Four-way ANOVAs for iCoh values between the PCC and ACC indicated the following results. There was a significant group × frequency × directionality × laterality interaction for iCoh values between the PCC and ACC connectivity among the four-way ANOVAs. Consequently, post-hoc independent *t*-tests for the iCoh ratio of directionality from PCC to ACC showed significant findings that the ratio was decreased in TRS compared with nTRS in delta (*t*_45_ = 7.659, *p* = 0.008; alpha = 0.01) and theta (*t*_45_ = 8.066, *p* = 0.007; alpha = 0.01) frequency bands in the left side (Figure 1). The results of ANOVAs and post-hoc independent *t*-tests are summarized in Appendix A.

Of note, there were no significant correlations between CPZ equivalent daily doses and iCoh values in either TRS group (r = −0.196, *p* = 0.225) or nTRS group (r = 0.064, *p* = 0.368).

### 3.3. Clinical Correlation with iCoh

Pearson’s correlational analyses indicated a trend toward a significant correlation between the iCoh ratio for the left delta PCC–ACC connectivity and PANSS total score only for TRS group (r = 0.38, *p* = 0.069), but not for nTRS group (r = −0.18, *p* = 0.17) (Figure 2).

### 3.4. ROC Analysis of the iCoh Ratio between TRS and nTRS

Regarding the discrimination between TRS and nTRS groups, the ROC analysis that employed the iCoh ratio for the left delta PCC–ACC connectivity showed a significant asymptotic *p*-value (*p* = 0.023; confidence interval: 0.536–0.868) with an area under the curve of 0.70. Further, the sensitivity and specificity at the optimum point of the ROC curve were 0.64 and 0.70, respectively.

## 4. Discussion

In the present study, we found that patients with TRS showed a significantly lower iCoh ratio between the PCC and ACC in delta and theta frequency bands over the left side than that of patients with nTRS. Furthermore, there was a trend toward a positive correlation between the PCC and ACC iCoh ratio in delta band over the left side and PANSS total scores in patients with TRS, but not in nTRS, suggesting that the higher iCoh ratio between the PCC and ACC in delta band over the left side was associated with greater psychotic symptoms severity in TRS group. These findings suggest that the absolute flow of information from the DMN to ACC was significantly attenuated in patients with TRS compared with patients with nTRS, while patients with TRS who had more severe psychiatric symptoms showed an increasing trend in relative information flow from the DMN to ACC.

Counterintuitively, we found a positive relationship between symptom severity and PCC–ACC iCoh ratio in the TRS group, while there was no association between them in the non-TRS group. These findings suggest that TRS may be accounted for by the hybrid model of categorical and continuous characteristics [32]. Although previous studies have shown consistent findings of the ACC abnormalities in patients with schizophrenia, no studies so far have examined the information flow in the cingulate bundle between the ACC and PCC in detail in this disorder. In addition, our result of a positive relationship between the PCC–ACC iCoh ratio and clinical severity may be related to impaired function of the ACC in patients with TRS.

The PCC is considered to play a crucial role in mediating spontaneous activity [33,34]. In addition, the PCC is thought to contribute to the essential functions such as emotional salience [35] and autobiographical memory [36]. Thus, dysfunction of the PCC may be related to clinical symptoms of schizophrenia such as hallucinations [37], delusions [38], or disorganized thinking [39]. Additionally, the PCC is also one of the core nodes of the theory of mind (ToM) network, which represents the cognitive ability to understand others as intentional agents by inferring their mental states [40,41]. Some studies showed that patients with schizophrenia had decreased activity of the ToM network including the PCC during the ToM task [5,42]. Thus, the PCC may be crucial as a pathological basis for this disorder. On the other hand, the ACC plays a role in mediating awareness and attention [43,44]. The ACC is thought to be a core region affected by schizophrenia [44] and dysfunction of the ACC may induce severity of symptoms or global impairment of cognitive function [45,46,47]. For example, a smaller volume of the ACC is significantly correlated with more severe positive symptoms of schizophrenia [45] and related to global cognitive impairment measured by the Brief Assessment of Cognition in Schizophrenia [47]. Notably, previous studies noted that patients with TRS had elevated levels of glutamatergic neurometabolites in the ACC [9,10,11]. These findings suggest that our results may explain the features of TRS including more severe positive symptoms, poorer cognitive function, and social function compared with nTRS. Additionally, impaired functional communication between the two regions might make them worse reciprocally. However, the ROC analysis using the iCoh ratio demonstrated a moderate accuracy to differentiate between patients with TRS and nTRS. While the iCoh ratio between the PCC and ACC may be a potential biomarker to distinguish between the two groups, future work is needed to disentangle the pathophysiology of TRS with a combination of multimodal biological measures.

Several studies have shown that the left cingulum is more related to positive symptoms of schizophrenia compared with the right cingulum. Reduced extracellular free-water as measured by diffusion MRI in the left cingulum was associated with delusions in patients with schizophrenia [48]. In addition, Palaniyappan et al. demonstrated that a reduced magnetization transfer ratio in the left cingulum was associated with a higher severity of delusions, while no such relationship was observed in the right cingulum [24]. Collectively, both functional and structural connectivities between the left PCC and ACC may be related to the severity of symptoms as represented by delusions. Additionally, Yuan et al. revealed that patients with schizophrenia who had never been treated for a long term showed more sever white matter abnormalities in the left cingulum-hippocampus pathway compared with patients with schizophrenia who had been treated [49]. This finding supports our hypothesis that the persistent symptoms observed in patients with TRS may be associated with functional abnormalities in the left cingulate cortex. Thus, these findings are in line with our result that the reduction of iCoh ratio was present only in the left side.

Unlike the neuroimaging studies, EEG enables the assessment of cortical network dynamics because of the high temporal resolution. Delta band oscillations are linked with learning, memory encoding and retrieval, and motivation and reward processes [50,51]. The activity of theta band oscillations has been linked to working memory, emotional arousal, and fear conditioning [51]. For example, Hlinka et al. reported that, in an inter-subject experimental design, a strong relationship was established between functional connectivity in delta band oscillations and the DMN [52]. Furthermore, Neuner et al. demonstrated a highly significant correlation between delta band oscillation and spontaneous blood-oxygen-level dependent (BOLD) signal within the DMN using simultaneous fMRI–EEG study [53]. Thus, delta band oscillations may represent the normal functioning of the DMN.

Our findings also suggest a new treatment option for TRS such as neuromodulation. Specifically, non-invasive novel neurostimulation techniques including transcranial magnetic stimulation (TMS) and deep brain stimulation (DBS) enable us to modulate the local neural connectivity [54,55]. Thus, the abnormal neural connectivity in the cingulate bundle can be one of the therapeutic targets. Given the limited treatment options for TRS, neurostimulation targeting the pathological neural basis as described above may be a promising therapeutic strategy.

There are several limitations to the present study. First, we did not include a healthy control group. Comparison between patients with schizophrenia and health control group may reveal comprehensive dysfunction in patients with schizophrenia, which will help to clarify the position of our current findings. Second, we did not include the potential covariates in the statistical analyses such as the dose of antipsychotics as indexed with CPZ equivalent daily doses. However, we did not see a correlation between the dose of antipsychotics and iCoh values or clinical severity. Third, the present study included relatively small sample sizes in both subdiagnostic groups (i.e., TRS and nTRS). Owing to Coronavirus disease 2019 (COVID-19), we could not continue to enroll subjects at this stage. Therefore, our findings warrant further studies with larger sample sizes in this illness using the TRRIP working group consensus criteria [28]. Forth, we focused only on the effective connectivity between the ACC and PCC as a hypothesis-based manner; however, there may be other potential network abnormalities in patients with schizophrenia. Further research is needed using multimodal imaging based on multifaceted perspectives. Fifth, we have only assessed psychiatric symptoms using the PANSS. Thus, there were no clinical measures for other symptoms like depression or anxiety. As schizophrenia is a multifaceted disorder, future research needs to include a variety of clinical measures. Lastly, in this study, we used the standard 10–20 EEG system using 19 electrodes, which could lead to incorrect localization. Because the number of source-level electrodes in the present study was small, we may not have been able to accurately estimate the source of the deep brain signals. Therefore, the present preliminary analyses warrant further studies to confirm the reproducibility and reliability of these results by, for example, using a higher resolution EEG system with 64 channel electrodes and combining them with more sophisticated signal source analysis techniques. However, several previous studies have performed the sLORETA analysis on 19-electrode EEGs [56,57,58,59].

## 5. Conclusions

In conclusion, we found significant differences in the iCoh ratio between the left PCC and ACC in delta and theta bands between patients with TRS and nTRS. Taken together, these findings may represent part of the underlying neural basis of TRS. The present findings warrant further research in larger sample sizes with multimodal examinations to elucidate underlying mechanisms of treatment-resistance in this illness.

## Figures and Tables

**Figure 1 jpm-10-00089-f001:**
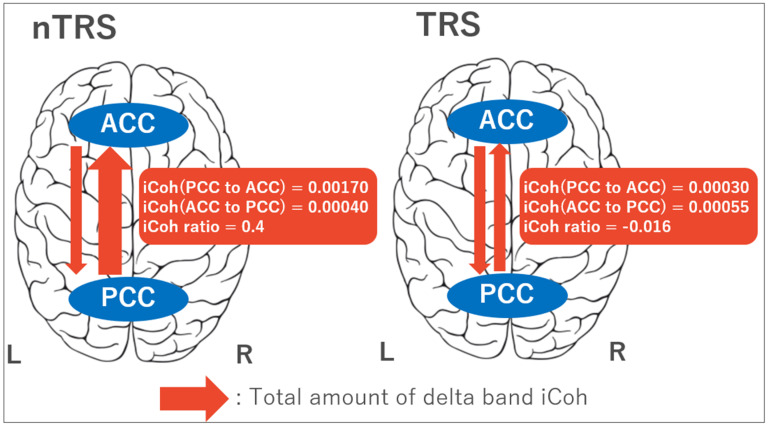
Schematics of the causal effective connectivity between the posterior cingulate cortex (PCC) and anterior cingulate cortex (ACC). In patients with treatment-resistant schizophrenia (TRS), the isolated effective coherence (iCoh) ratios [(PCC to ACC) − (ACC to PCC)]/[(PCC to ACC) + (ACC to PCC)] in delta and theta bands over the left side were significantly decreased compared with patients with non-TRS (nTRS).

**Figure 2 jpm-10-00089-f002:**
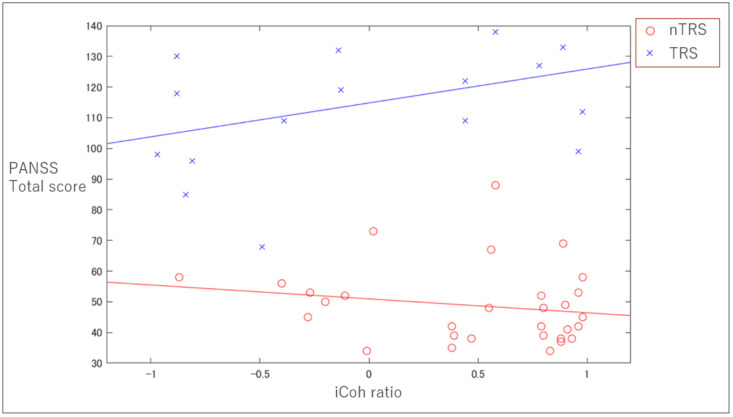
The relationship between the iCho ratio for the left delta PCC–ACC connectivity and positive and negative syndrome scale (PANSS) total score. In patients with TRS, there is trend toward a relationship (r = 0.38, *p* = 0.069), but not for the nTRS group (r = −0.18, *p* = 0.17).

**Table 1 jpm-10-00089-t001:** Clinico-demographics data.

	nTRS (n = 30)	TRS (n = 17)	*t*-Value (Chi-Squared Value for Sex), *p*-Value
Age, mean (SD), years	41.2 (12.6)	42.4 (13.4)	*t*_45_ = 0.29, *p* = 0.78
Sex (number of male) (%)	13 (43)	5 (29)	χ^2^_45_ = 0.89, *p* = 0.34
Education, mean (SD), years	13.3 (1.81)	13.3 (2.42)	*t*_45_ = 0.06, *p* = 0.95
Age of onset, mean (SD), years	26.0 (9.47)	26.6 (7.64)	*t*_45_ = 0.23, *p* = 0.82
Treatment duration, mean (SD), years	14.5 (12.0)	15.5 (11.3)	*t*_45_ = 0.27, *p* = 0.79
Chlorpromazine equivalents, mean (SD), mg	406 (233.5)	748 (319.0)	*t*_45_ = 4.22, *p* < 0.001 *
PANSS total, mean (SD)	48.8 (12.7)	114.6 (21.2)	*t*_45_ = 13.4, *p* < 0.001 *

TRS: treatment-resistant schizophrenia; nTRS: non treatment-resistant schizophrenia; SD: standard deviation; PANSS: positive and negative syndrome scale, * = *p* < 0.05.

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
