# Peer review of "Resting-State Isolated Effective Connectivity of the Cingulate Cortex as a Neurophysiological Biomarker in Patients with Severe Treatment-Resistant Schizophrenia"

_jpm, 2020, doi:10.3390/jpm10030089_

Round 1

Reviewer 1 Report

The authors studied the effective connectivity between the posterior (PCC) and anterior (ACC) cingulate cortices in treatment resistant schizophrenia as compared to the non-treatment resistant schizoprenia patients. To this end, they recorded resting-state scalp EEG with 19 electrodes arranged according to the 10-20 international system and calculated the isolated effective coherence (iCoh) between the two regions of interest within each hemisphere. There is a great interest in the effective connectivity in schizophrenia. There are, however, number of weaknesses in this study:

1) It has been shown previously that routine clinical EEG recorded with the standard 10-20 system using less than 32 electrodes leads to incorrect localization (Michel and He, 2012; Michel et al. 2004; Srinivasan et al., 1998). The spatial resolution of the 32-channel array was estimated to about 7 cm, i.e. approximatelly to the size of a lobe of the brain (Srinivasan et al., 1998).Thus, a correct source localization made with the 19-electrode array is not possible and as such precludes any analysis of connectivity. I refer authors to a recent pracital review on EEG source imaging to improve their study (Michel and Brunet, 2019).

2)The authors did not sufficiently describe how the source localization was made. They should for example report, whether they used generic or individual head models for the source reconstruction.

3) It would be also interesting to  assess the effective connectivity properties of other regions of interest within for example the default mode or saliency networks, since in schizophrenia not only the here reported specific path might be impaired.

Minor comments

1) The title is not clear. Did you mean some „correlation with“ or some „correlates of“?  If I understood it correctly, the authors studied the resting-state iCoh between the ACC and PCC as a neurophysiological correlate of treatment resistance in schizophrenia.

2) Please unify the abbreviations TnRS, nTRS

3) Fig 1: all abbreviations used in the figure should be explained in the legend. The schematic relation between the PANSS and connectivity results adds no additional information with respect to the reported results in the text and could be deleted from the figure.

4)Table 1: all abbreviations should be explained, on the other hand CPZ is not used in the table, therefore need not be explained.

5)Author Contributions: It is very nonstandard that the author (FM), who did the analysis, visualisation and wrote the intial draft is not the first author of the manuscript. In my opinion all authors should review and edit the manuscript. On the other hand those who just reviewed and edited the manuscript should be mentioned in Acknowledgement rather than being co-authors. The terms „Investigation“ and „Resources“ are vague and difficult to understand.

Michel, C. M., Murray, M. M., Lantz, G., Gonzalez, S., Spinelli, L., & Grave De Peralta, R. (2004). EEG source imaging. Clinical Neurophysiology, 115(10), 2195-2222. doi:10.1016/j.clinph.2004.06.001

Michel, C. M., & Brunet, D. (2019). EEG source imaging: A practical review of the analysis steps. Frontiers in Neurology, 10(APR) doi:10.3389/fneur.2019.00325

Michel, C. M., & He, B. (2012). EEG mapping and source imaging. Niedermeyer's electroencephalography: Basic principles, clinical applications, and related fields: Sixth edition (pp. 1179-1202)

Srinivasan, R., Tucker, D. M., & Murias, M. (1998). Estimating the spatial nyquist of the human EEG. Behavior Research Methods, Instruments, and Computers, 30(1), 8-19. doi:10.3758/BF03209412

Reviewer 2 Report

The authors present some interesting findings concerning the neural mechanisms underlying treatment-resistant schizophrenia. They use a novel resisting-state EEG connectivity approach (iCoh) to compare effective connectivity between ACC-PCC, PCC-ACC between two groups of patients with schizophrenia categorised as treatment-responsive and treatment-resistant. The authors report a significant decrease in delta and theta bands in the connection from PCC-ACC on the left side in TRS patients compared to nTRS, and left delta connectivity was weakly negatively related to total PANSS symptoms in the TRS group only. This suggests that information flow from PCC-ACC in TRS is attenuated and is related to increase symptom severity in this group – a possible future treatment target. On the whole, I found the paper to be well-written, concise and interesting but there were a few points that needed further explanation or clarification:

Intro

  1. “Thus, dysfunction of the ACC may not only be a pathological neural basis for schizophrenia in general but also be related to the pathophysiological mechanisms underlying TRS.” (lines 70/71). I am not quite sure this makes sense – could the authors clarify what pathological and pathophysiological mechanisms they are referring to?
  2. The authors talk about altered connectivity within the DMN in schizophrenia and that this relates to symptoms (lines 72 -83). This is great but there is no much explanation of how this relates to symptoms. Can the authors expand this paragraph to clarify why the DMN might be related to symptoms? What is the DMN important for and why might altered connectivity be apparent in schizophrenia?
  3. The authors give a brief description of iCoh in the introduction (lines 84-89). I realise that this is described more in the methods section but since this is a new method (one I have not come across before), it would be nice to have a fuller description of what this is, how it can be used and what makes it different from other methods that could have been used – rationale.

Methods

  1. Line 97 – please add the city/country of Komagino Hospital and where the study took place. Also ethical approvals and informed consent (requirement of the journal?).
  2. Also state that the clinic-demographic data were compared between groups using t-tests etc.
  3. Line 121 – include reference for software.
  4. Line 136/7 – could the authors clarify what they mean by ‘hub pairs’. How were these chosen and defined? Below, the authors then talk about ROIs and these are listed in supp materials. Are the ROIs the same as the hub pairs? Again, how were they defined? Need a clearer description.

5 Line 152. The authors report they use an alpha level of 0.05. How were the p values adjusted to this level?

Results

  1. Table 1 looks good. Please clarify what the * symbol refers to.
  2. Could the authors clarify Figure 1. It’s not very clear what the arrows refer to i.e. are the delta and theta bands separate or combined together in the figure? If so, it is not labelled. Be clear that the thinner arrow represents decrease in ratio – perhaps the ratio value could be included in the figure? The relationship between PCC/ACC connectivity and PANSS in TRS is hinted at in the figure but not described in figure legend or referred to in text (i.e. lines 183-185). Could this be clarified. Preferably, it would be good to see the relationship too e.g. in scatter plot.

Discussion

Generally a good summary of the findings.

  1. However, discussion of the theory behind the results is a bit weak. For example, line 208-210 discussing glutamate disturbances is repeated from the introduction but it doesn’t really explain how the current findings support that study and how it adds to it. In general, I think the discussion could be improved with some further explanation of what the function of the PCC-ACC connection is and why it might be decreased in TRS. The PCC is generally considered one of the DMN’s hubs for switching attention between internal thoughts and external interactions whilst the authors note that ACC is important for cognition etc. There is robust evidence for poorer cognitive control, executive function, working memory etc in TRS and it’s thought that lack of cognitive control might explain the persistence of symptoms in these patients e.g. https://www.nature.com/articles/s41598-019-51023-0. Some discussion of these points are needed.
  2. Lines 219/220 – again, more discussion of how the PCC-ACC connection may underly persistent symptoms in TRS.
  3. Line 238 – the authors note no healthy control group in the limitations. This is good but a little brief – how does this limit the current findings?

Other:

A couple of typos: line 20 in abstract and Table 1 headings (TnRS).

Reviewer 3 Report

The manuscript explores the causal effective connectivity between the anterior cingulate cortex (ACC) and a brain area of the default mode network (DMN), that is the posterior cingulate cortex (PCC), in patients with treatment-resistant schizophrenia (TRS). To this end, resting-state EEG was acquired in both patients with TRS and in patients with non-TRS. The main result is that patients with TRS showed significantly lower iCoh ratio between the PCC and ACC in delta and theta frequency bands over the left side than that of patients with nTRS. The authors suggest that this may be a potential biomarker to distinguish between the two groups of patients.

Overall, the paper is worthy of attention and I consider the topic of this study to be potentially of interest to readers of the Journal of Personalized Medicine. Here some revisions I request before the manuscript can be considered publishable:

  1. In the conclusion section of the abstract the authors state that “Left delta PCC and ACC iCoh ratio may represent a neurophysiological basis of TRS.” Given the data shown, I wonder why the authors omit a reference to the results concerning the theta frequency band.
  2. In the abstract there is a typo. “we aimed to examine identify…”
  3. In the Introduction the authors state that “One brain region commonly reported to show abnormal structural and functional findings in patients with schizophrenia is the anterior cingulate cortex (ACC)”. Here references to the appropriate literature are necessary (e.g., PubMed ID: 19287044, 23049832, 11532726).
  4. What is the reason why the authors based their diagnoses of schizophrenia or schizoaffective disorder on the fourth instead of the fifth edition of the DSM?
  5. Lines 236-238: the authors state that “the PCC is thought to contribute to the essential functions such as emotional salience and autobiographical memory. Thus, dysfunction of the PCC may be related to clinical symptoms of schizophrenia such as hallucinations, delusions, or disorganized thinking”. Here it is import that the authors recognize that the PCC is also essential for a further key function, that is “Theory of Mind” (e.g., PubMed ID: 28438665, 31773921) and, thus, that dysfunction of the PCC may be related to the well-know altered mentalizing ability in schizophrenia (e.g., PubMed ID: 19287044, 8541250).
  6. Severity of clinical symptoms was assessed with the PANSS only. Because the authors used resting-state EEG to examine the connectivity between the ACC and PCC (a hub of the DMN) this should be problematic. In fact, Imperatori et al. (2019, PubMed ID: 30605880) by means of EEG functional connectivity demonstrated DMN alterations during resting-state in both theta and beta bands in individuals with high-trait-anxiety. Considering the significant individual differences in anxiety levels among patients with schizophrenia, the assessment of anxiety should have required additional clinical measures, such as the State-Trait Anxiety Inventory (STAI). This must be recognized as a limit of the present work and/or as a suggestion for future work.

Round 2

Reviewer 1 Report

I appreciate the opportunity to evaluate the manuscript that I suggested to reject in my previous review. Following a careful assessment of the paper I see the hard work authors made that improved their manuscript in minor concerns that I raised. Nevertheless, the main weakness of the study, i. e. the insufficient number of electrodes to study effective connectivity between the ACC and PCC, was not properly addressed.

As I pointed out in my previous review, the even cortical sources can be observed with the spatial resolution of 7 cm in case of the 32-channel array (Srinivasan et al., 1998). For the deeper cortical sources such as the ACC and PCC, what´s more, using only 19-channel array, the spatial resolution gets much worse to almost impossible. Furthermore, the connectivity measures cannot be assigned to these structures because of that spatial 'blur' given by the low-density recordings.

The authors argue that 19-channel array was used in previous source-localizing studies. This is, however, a weak argument, if any. In all these studies the spatial resolution of the source reconstruction was not crucial for interpretation of their results. In contrast to the current study, the recently reviewed (Coben et al., 2019) neurofeedback studies with 19-channel arrays had no ambition to investigate the function of the brain structures per se and rather used the source reconstructed signal as a feedback to learn volitional control over this signal. Similarly, the case-study finding of increased source reconstructed neuronal activation from baseline in the patient following 6-weeks of treatment, in my opinion, cannot be taken as a proof of precise source-localization performed with 19-channel array(Eugene et al. 2016). Such treatment-related increase might have been observed even if the source reconstruction was inacurrate. Concerning the study by Ponomarev and Kropotov (2013), bluring effect of the low-density EEG is clearly visible in the data presented, e.g. in the independent component No. 2 in Fig.3b, demonstrating how problematic the assignement of the observed activity to a specific brain structure might be, when a low spatial resolution EEG is used. The study on effect of meditation on brain activity by Lehmann et al. (2012)  that was also performed on 19-electrodes, assessed, however, the change of global functional interdependence between brain regions and did not assign the observations to any specific brain structure.

I hope that my comments will be of value to the authors, and contribute to the progress of their project as well as encourage them to pursue further their ideas. To investigate the effective interdependence between the ACC and PCC I suggest repeating the study on high-density recordings preferably employing individual head models for the source reconstructions.

References:

Coben, R., Hammond, D. C., & Arns, M. (2019). 19 channel Z-score and LORETA neurofeedback: Does the evidence support the hype? Applied Psychophysiology Biofeedback, 44(1) doi:10.1007/s10484-018-9420-6

Eugene, A.R., Masiak, J. Identifying treatment response of sertraline in a teenager with selective mutism using electrophysiological neuromaging. Int J Clin Pharmacol Toxicol 2016 Jun;5(4):216-219

Lehmann, D., Faber, P. L., Tei, S., Pascual-Marqui, R. D., Milz, P., & Kochi, K. (2012). Reduced functional connectivity between cortical sources in five meditation traditions detected with lagged coherence using EEG tomography. NeuroImage, 60(2), 1574-1586. doi:10.1016/j.neuroimage.2012.01.042

Ponomarev, V. A., & Kropotov, Y. D. (2013). Improving source localization of event-related potentials in the GO/NOGO task by modeling their cross-covariance structure. Human Physiology, 39(1), 27-39. doi:10.1134/S036211971301012X

Srinivasan, R., Tucker, D. M., & Murias, M. (1998). Estimating the spatial nyquist of the human EEG. Behavior Research Methods, Instruments, and Computers, 30(1), 8-19. doi:10.3758/BF03209412
